# (Bio)printing in Personalized Medicine—Opportunities and Potential Benefits

**DOI:** 10.3390/bioengineering10030287

**Published:** 2023-02-23

**Authors:** Dobromira Shopova, Antoniya Yaneva, Desislava Bakova, Anna Mihaylova, Petya Kasnakova, Maria Hristozova, Yordan Sbirkov, Victoria Sarafian, Mariya Semerdzhieva

**Affiliations:** 1Department of Prosthetic Dentistry, Faculty of Dental Medicine, Medical University, 4000 Plovdiv, Bulgaria; 2Department of Medical Informatics, Biostatistics and eLearning, Faculty of Public Health, Medical University, 4000 Plovdiv, Bulgaria; 3Department of Healthcare Management, Faculty of Public Health, Medical University, 4000 Plovdiv, Bulgaria; 4Department of Medical Biology, Medical University, 4000 Plovdiv, Bulgaria; 5Research Institute, Medical University, 4000 Plovdiv, Bulgaria

**Keywords:** 3D (bio)printing, opportunities, systematic review, personalized medicine

## Abstract

The global development of technologies now enters areas related to human health, with a transition from conventional to personalized medicine that is based to a significant extent on (bio)printing. The goal of this article is to review some of the published scientific literature and to highlight the importance and potential benefits of using 3D (bio)printing techniques in contemporary personalized medicine and also to offer future perspectives in this research field. The article is prepared according to the Preferred Reporting Items for Systematic Reviews and Meta-Analyses (PRISMA) guidelines. Web of Science, PubMed, Scopus, Google Scholar, and ScienceDirect databases were used in the literature search. Six authors independently performed the search, study selection, and data extraction. This review focuses on 3D bio(printing) in personalized medicine and provides a classification of 3D bio(printing) benefits in several categories: overcoming the shortage of organs for transplantation, elimination of problems due to the difference between sexes in organ transplantation, reducing the cases of rejection of transplanted organs, enhancing the survival of patients with transplantation, drug research and development, elimination of genetic/congenital defects in tissues and organs, and surgery planning and medical training for young doctors. In particular, we highlight the benefits of each 3D bio(printing) applications included along with the associated scientific reports from recent literature. In addition, we present an overview of some of the challenges that need to be overcome in the applications of 3D bioprinting in personalized medicine. The reviewed articles lead to the conclusion that bioprinting may be adopted as a revolution in the development of personalized, medicine and it has a huge potential in the near future to become a gold standard in future healthcare in the world.

## 1. Introduction

Three-dimensional (3D) bioprinting has gained increasing interest and has been widely used in the healthcare field in recent years [1].

The global development of technologies and their entry into areas related to human health has led to a certain transformation of the model of conventional medicine towards the application of a more personalized approach. Along with other medical biotechnologies, personalized medicine is being developed and complemented to a great extent with the application of (bio)printing [2,3,4]. Recently, the implementation of these technologies has grown rapidly and is expected to cause a real revolution in the field of health care [5,6]. In 2006, Mironov et al. defined bioprinting as: “The use of material transfer processes for patterning and assembling biologically relevant materials—molecules, cells, tissues, and biodegradable biomaterials—with a prescribed organization to accomplish one or more biological functions” [7]. Groll et al. reappraised the definition: “bioprinting refers to the use of computer-aided transfer processes for patterning and assembling living and non-living materials with a prescribed 2D or 3D organization to produce bio-engineered structures” [8]. The three-dimensional creation of objects from biocompatible materials is not enough to claim that bioprinting has taken place. It is also described as an innovative business model. It can be used to create skin, blood vessels, and cartilaginous structures all the way up to whole organs—heart, liver [9]—as well as for the fabrication of wearable sensors for health monitoring that can gather real-time data and facilitate disease diagnosis [10,11]. Some prototypes are applied to drug testing, toxicity in tumour treatment, etc. Human anatomical models are also created and used to teach medical students during their studies. As a future goal, the thesis of regenerating organs and organs that grow with the growth of the organism is also being developed, which is especially important in neonatology [12,13].

Tissue or organ failure due to aging, disease, accidents, and congenital defects is a serious medical problem, the solution of which depends primarily on organ transplants from living or deceased donors. There is a worldwide shortage of human organs for transplantation as the number of patients in need continues to rise. The percentage of those who survive to wait for an organ is extremely low. An additional difficulty arises from the fact that organ transplantation is often associated with the rejection of the transplant. This problem could be eliminated by using genetic material from the patient’s body to create a replacement organ as a transplant. This would minimize the risk of rejection and the need to take lifelong immunosuppressive drugs [14,15]. Therapies based on tissue engineering and regenerative medicine are being reported as a potential solution to the shortage of organ donors. The strategy is to isolate stem cells from small tissue samples and stimulate them with growth factors that direct cell proliferation and differentiation into functioning tissues (Figure 1). So-called bioprinting is a particular form of medical printing. In this case, special products (so-called “bioinks”) are applied, which can be printed using three-dimensional methods. Bioinks consist of a natural hydrogel such as collagen, gelatin, or synthetically produced hydrogels and are loaded with live cells. In this way, the desired tissue architecture and shapes can be 3D printed. The goal is for the single cells themselves to be self-assembled into cell complexes and tissues. Therefore, it is necessary to use the expertise of scientists who have been actively working in the field of tissue engineering in recent years. The focus of current scientific developments and research is directed towards the creation of mainly soft tissues [16,17,18].

In order to overcome the demographic challenges, the increasing frequency of degenerative diseases, and the growing number of patients, the expanding socio-medical significance of (bio)printing is being considered. The creation of individual patient-specific implants and truly original printed models demonstrates this technology’s enormous potential, which could significantly improve patient care through further research. The pre-planning and visualization of the surgical treatment by 3D (bio)printing have been established in clinical practice, especially in cases of unusual anatomical deformities [20,21]. In addition, these printed models could be used to pre-inform the patient about upcoming interventions.

Thanks to the excellent results, the production and demand for printed medical products will continue to grow in the future. The main advantage is related to the high degree of product identity with the patient’s body, which improves the adaptation process, especially in the case of anatomical deformities. This, in turn, will lead to faster healing and recovery [22]. Another area of application of bioprinting is the creation of tissues and cells that pharmaceutical companies and researchers can use to test new drugs on [23,24]. This revolutionary advance will greatly aid personalized medicine [3,25]. 

Tissue engineering will help create cell-containing structures in a computer-controlled manner, thus sparing costly and insufficiently controlled manual cell seeding [26]. Biomaterial development should primarily focus on and deal with some key issues (conforming degradation to tissue development and ensuring adequate mechanical properties while obtaining rheological properties needed for the manufacturing process), structure design (including vascularization of the structure), and system integration (integration of multiple cells, materials, and manufacturing processes in a sterile and controlled environment) [27,28,29]. The major prospect of bioprinting is the ability to print and pattern all the components that comprise a tissue (cells and matrix materials) in three dimensions to generate structures similar to tissues [30]. Also, printing tissue analogue constructs is vital for some emerging technologies that require the delivery of living cells with appropriate material in a defined and organized manner, at the right location, in sufficient numbers, and within the right environment [29,31,32]. The 3D-bioprinted organ will preserve the structural, mechanical, biological, and metabolic properties as those of a normal and healthy organ [33]. 

Tissue-engineering scaffolds, cell-based sensors, drug/toxicity screening, and tissue or tumour models are some of the printing technologies [34,35]. An alternative method of ink-jet printing and bioplotting in the assembly and micropatterning of biomaterials and cells comes from biological laser printing (BioLP) using laser-induced forward transfer. High-efficiency biological laser printers have been developed with the capacity to deposit a wide range of biological components, all of which are required for tissue engineering: biopolymers, nano-sized particles, and human endothelial cells. It highlights the criteria for building 3D structures using the bioprinting process: the writing/extrusion/polymerisation speed, the volume fraction of deposited materials, the process resolution, and its capacity to be integrated with other tissue engineering methods [1,36,37]. Bioprinting allows the manufacturing of scaffolds with an organized and homogeneous distribution of different cell types within and throughout the construct, simulating tissues with multiple cell types or the interface between two tissues. Not only does the choice of material and design impact the viability and proliferation of the printed cells, but the different techniques have also shown variable cell activities post-production [38,39].

The purpose of this article is to review some of the published scientific literature and to highlight the importance and potential benefits of using 3D (bio)printing techniques in contemporary personalized medicine, and, also, to outline the future perspectives in this research area. As a fast-developing field, an extraordinary interest of scientists in bioprinting and its applications has been registered. Our analyses have shown that in the last 5 years, there is an exponentially growing number of scientific publications on this topic. Therefore, it is of great importance for the awareness and understanding of this field to review these studies and to systematize both the areas of application and the potential benefits and challenges of the implementation of bioprinting methods and techniques. All of them possess the potential to lead to revolutionary changes in medicine as well as to influence many aspects related to personalized therapy. In addition, this article discusses the use of 3D printing techniques and points to the possible challenges in this research field, with the aim to propose guidelines for future research.

## 2. Materials and Methods

The Preferred Reporting Items for Systematic Reviews and Meta-Analyses (PRISMA) guidelines were followed to perform this work [40].

### 2.1. Literature Search

The authors systematically and comprehensively saved all studies on the development and application of 3D printing in the medical/healthcare field from the Web of Science, PubMed, Scopus, Google Scholar, and Science Direct databases. Search terms ((3D bioprinting) OR (bioprinting)) AND (personalized medicine) were used to identify articles related to personalized medicine. Search terms ((3D bioprinting) OR (bioprinting)) AND ((cells) OR (tissue) OR (organ)) were used to identify articles, related to cells, tissue, and organ bioprinting applications. Search terms ((3D bioprinting) OR (bioprinting)) AND ((opportunities) OR (advantages)) AND ((review) OR (systematic review)) were used to identify articles describing and outlining some advantages.

### 2.2. Eligibility Criteria

The inclusion criteria comprise articles published between 2017 and 2022, reviews, systematic reviews or meta-analyses, and full-text articles. The exclusion criteria include abstracts, short communications, patents and policy makers, case reports, and studies in which fundamental information about the additive manufacturing process and bioprinting is missing. 

We incorporated all articles regardless of language restrictions and produced a results summary.

### 2.3. Data Analysis

We used Microsoft Office Excel 2010 as a tool to develop a data extraction form with the intention to standardize the extraction and analysis of data. The articles were selected from the databases and organized in an Excel file, eliminating the repeating ones. After that, all the abstracts of the articles were examined separately by three authors, from which they chose several papers, the full texts of which were again independently read, thereby allowing us to reach a final selection of relevant articles. Few experimental research and prospective studies comply with all the inclusion and exclusion criteria, as it is known that studies on the implementation of 3D and bioprinting are relatively new. Several sources before 2017 were also a part of this literature view. Having finished a precise literature selection, the choices of all authors were considered in parallel and discussed until a final selection was achieved. 

## 3. Results and Discussion

Out of the five selected databases, 714 potentially suitable articles were sorted. An overall amount of 482 studies were included after eliminating the repeating ones. After reading the abstracts, 263 works were dropped as they did not contain sufficient data and/or due to different study strategies. As a result, 219 complete articles were analysed, and then only 120 full-text articles were selected. Figure 2 demonstrates the PRISMA flow chart for the selection process of articles included in this systematic review.

After a detailed review and discussion of the synthesized scientific literature, we arrived at the most significant identified benefits of the application of bioprinting for the development of personalized medicine. Thereafter, three domains and five subdomains, namely, tissue and organ transplantation, drug research and development, and surgical planning, were considered for the review of the potential benefits of 3D bioprinting in the personalized healthcare process/sector (Table 1).

### 3.1. Benefits of Organ Transplantation

#### 3.1.1. Transplant Organ Shortage

It is indisputable that the lack of organs is reaching a critical point [41]. The overall number of patients on the waiting list greatly exceeds the total number of available donors. Furthermore, the successful completion of transplantation consists both of receiving a healthy organ and coping with challenges in combating infections and tissue rejections [42,82]. The Global Donation and Transplantation Observatory shared data for 129,681 organ transplants internationally in 2020, which is 17.6% less than in 2019, when a total of 157,301 organs were transplanted [83]. According to organ donation statistics, about 20 patients die every day while waiting to receive an organ transplant offer [43]. Until now, organs are provided by cadavers and living donors. Donor campaigns have a very important role, and the ones who urge for organ donations save and have saved lives. However, if the sale of organs becomes legalized, donors would be less as the moral significance of donating an organ would be diminished [44]. 3D printing can attribute to transplantations and thus critically enhance the quality of life for many people around the world [45].

#### 3.1.2. Overcoming Gender Differences in Transplantation

Clinical processes, treatment opportunities, and outcomes of organ donation and transplantation depend on sex and gender. Sex and gender differences in kidney function, anatomy, and physiology have been noticed, and they have an impact on kidney donation and transplantation due to differences in kidney size (sex aspect) and altruism (gender aspect). Therefore, by providing personalized medicine, 3D bioprinting might be anticipated to remove sex and gender influence [84].

Of note, immunosuppressive drugs lead to different responses in men and women. In addition, eventually, the graft survival might be affected by some infections that are very common after transplantation, such as HIV, BK virus, and tuberculosis. Male sex is a risk factor for BK virus infection according to a study by Momper et al. [46]. Vnucak et al. discovered major sex differences in the frequency of single and repeat infections in different time intervals after kidney transplantation. The female sex is a risk factor for the incidence of infection generally [47]. Post-transplant mortality also depends on sex and gender. The largest three transplant databases worldwide have been used by Vinson et al. to analyse 438,585 patients with transplantation between 1988 and 2020: the American SRTR, the Australia and New Zealand Dialysis and Transplant (ANZDATA) Registry, and the international Collaborative Transplant Study (CTS) database. A connection was found between recipient sex and kidney transplantation survival that varies by recipient age and donor sex [48].

Sex and gender have a strong effect on the medical process of liver transplantation. Research by Serrano et al. studies recipient sex as a risk factor for death of patients after liver transplantation by using survival analysis. Among patients, men show lower short-term mortality than women but higher long-term and overall mortality [49]. A study by Gabbay et al. makes an assessment of sex-specific outcomes after liver transplantation, particularly short-term mortality and long-term survival rates. The authors established better female outcomes, i.e., lower short-term mortality and higher long-term survival [50]. 

In order to better understand the molecular mechanisms of disease development and the impact of drugs on nephrotoxicity, scientists have been creating and testing more physiologically representative models of human organs [85,86]. It is highly relevant to anticipate that 3D bioprinting, being a form of personalized medicine, might eliminate sex and gender differences. Therefore, sex and gender should be considered in every stage of the process of 3D bioprinting. Sex and gender affect clinical processes, opportunities for treatment, and consequences of organ donation and transplantation [84]. 

#### 3.1.3. Probability of Reduction of Transplant Rejection

This research examines the advancement in 3D bioprinting of vital organs and basic immune response against the biomaterials used in this process. Assessing immune reactions to biomaterials used in 3D printed organs is required in order to mitigate tissue rejection after transplantation [87,88]. The higher immunotolerance of the biological cells used in bioprinting has a significant beneficial effect on the transplantation process, thus leading to faster healing and fewer complications after the transplantation [89,90]. The recent progress of three-dimensional printing of biomaterials in personalized medicine has been described to have the potential to change some medical treatments, including achieving new responses to organ damage or organ failure [91]. 

When the immune system becomes suppressed, for example, after kidney transplantation, a virus can appear or reappear from a previous infection in the kidney, thus ultimately leading to the rejection of the organ [51]. Female donor kidneys tend to be rejected more often after transplantation than male donor kidneys, which is due to the difference in kidney volume. Poorer kidney survival is linked to the reduced number of nephrons in smaller kidneys [58]. Additionally, despite the fact that larger persons can receive smaller organs, smaller persons cannot receive larger organs [92].

In the future, personalized tissues and organs will be created (printed) for patients in need, which will minimize the possibility of their rejection. In this way, overall organ recovery and tissue regeneration will be significantly improved [93]. In addition, patients needing organ transplantation may choose to have live, lab-grown kidneys and other organs instead of the risk of waiting hopelessly for years in donor-organ queues [52]. As a result of the joint work of researchers, it would be possible to fabricate biomimetic patient-specific tissues/organs. Scientists are heading toward safer, healthier major organ replacements with three-dimensional (3D) bioprinting [88] that will increase patients’ chances of survival.

Traditional methods and materials used nowadays to treat some diseases show disadvantages, which is anticipated for almost every treatment method. Knowing this, safe and efficient alternatives are required, particularly in medical cases where conventional methods do not prove to be as beneficial to the patient [53]. It is here that the potential of 3D and bioprinting as effective alternatives emerges [94]. 

The wide range of applications of 3D printing is well known in the world of engineering, and it is wonderful to see the emergence of these technologies in medicine thanks to medical engineering. Bioprinting has been implemented in different ways in regenerative medicine in orthopaedics, plastics/dermatology, cardiovascular surgery, etc. [54,55,57,95]. Therefore, it can be concluded that 3D bioprinting offers significant potential as an efficient and safe alternative to the traditional methods and materials [96]. This method will help increase the rate of successful implantation of mature tissues into patients. This approach could enhance the integration of the implanted tissues into the patient’s native myocardial tissues, thereby restoring cardiac functions after a cardiac injury [58,97].

Urgency is the major reason for the focus on kidney transplantation: currently, kidneys are the most frequently transplanted organs worldwide. The availability of 3D-bioprinted kidneys would result in fewer kidney patients dependent on donor programs to find a matching donor, likely making 3D bioprinting widely implemented once available [98,99]. Another severe traumatic injury is burning. In the course of the years, the main patient care has transformed from just survival to helping and providing improved functional outcomes. Usually, the treatment of burns, especially in the case of extensive ones, comprises surgical excision of injured skin and reconstruction with skin substitutes. Traditional skin substitutes do not possess all skin cell types and thus do not help with the recapitulation of native skin physiology [100,101].

A constant uninterrupted and interrelated research flow should be deployed to achieve technological advancement along with corresponding epigenetic mechanisms with regards to producing new structures. Thus, 3D bioprinting technologies would become one of the most reliable, efficient, and favorable methods to fabricate tissue structures in the near future [59]. 

#### 3.1.4. Removal of Congenital Defects of Various Tissues and Organs

3D bioprinting, as opposed to non-biological printing, entails additional complexities, including the choice of materials, cell types, growth and differentiation factors, and technical challenges associated with the sensitivities of living cells and the construction of tissues. Finding a solution to these complexities demands the integration of technologies from the fields of engineering, biomaterials science, cell biology, physics, and medicine. 3D bioprinting has already been implemented for the generation and transplantation of several tissues, including multi-layered skin, bone, vascular grafts, tracheal splints, heart tissue, and cartilaginous structures [55]. 

A study by Lee et al. presents a 3D printing technique for producing complex collagen scaffolds for engineering biological tissues. Modulation of pH was used to control collagen gelation, and it could provide up to 10-micrometer resolution on printing. Cells could be embeded in the collagen or pores could be introduced into the scaffold via the use of gelatin spheres. In this study the authors showed a successful 3D printing of five components of the human heart spanning capillary to full-organ scale, which was validated for tissue and organ function [60].

Congenital tracheomalacia and tracheal stenosis are common cases among premature infants. In adulthood, they are usually linked with chronic obstructive pulmonary disease, and can occur secondarily from a tracheostomy, prolonged intubation, trauma, infection, and tumours. When these conditions are not controlled properly, they are life-threatening. We know that particular pathologies are still facing some surgical limitations, but here comes tissue engineering as a promising approach to treat significant airway dysfunctions. Currently, 3D-bioprinting has helped in preclinical and clinical efforts for airway reconstruction [61].

Having in mind the great necessity in the reconstructive surgery, 3D bioprinting presents itself as a promising technology that might be able to produce rapidly and reliably biomimetic cellular skin substitutes, to the satisfaction of both clinical and industrial needs [102]. Many of these prostheses may not integrate biologically properly and may not function biomechanically when transplanted. It has been demonstrated that a 3D printed tracheal scaffold could deliver an alternative solution as a therapeutic treatment for partial defects [62].

The emergence of new diseases and the increasing human population, suggest that over time the number of potential patients is anticipated to grow. Bioprinting might offer a solution to these issues, however many modifications and development are likely to be required in order to result in a living organ combining multiple cell types and materials [64].

### 3.2. Drug Research and Development

The implementation of tissue-specific models for bioprinting organs or specific tissues assists in the testing of therapeutic schemes and supports the clinical diagnosis and treatment of disease by replacing the injured tissues. It will be possible to determine beforehand personalized pathophysiological conditions considering the genome, proteins, and medical/family history, and therefore limiting life-threatening effects [103]. 

3D bioprinting provides exciting opportunities for printing 3D tissue/organ models. This technique can reproduce the spatial and chemical complexity characteristic of native tissues and organs. This is why bioprinted tissues/organs might be of great help in the prioritization of lead candidates, toxicity testing, and disease/tumour models. Bioprinted tissues/organs may facilitate the testing of novel compounds or predicting toxicity because the spatial and chemical complexity characteristic of native tissues/organs can be recreated [84]. Last but not least, additive manufacturing with regards to drug printing may also indicate an innovative technique in the production of patient-specific medicines and in personalizing the composition and the dose needed by the patients. Drug-printing raises the idea of tailor-made drugs, which makes them safer and more effective [104,105].

3D bioprinting can be implemented in target identification and validation in the process of drug discovery and development. 3D bioprinted disease models might be used for high-throughput screening for in vitro efficacy when it comes to the hits-to-leads step. Regarding leads optimization, 3D bioprinted constructs may be used for in vitro efficacy evaluation and retrospective toxicity assays after confirmation of target organs from in vivo toxicity assays. After continuously refining loops of structure-activity relations, synthesis of new compounds, and in vitro and in vivo analyses, candidate drugs are sorted from optimized leads after the regulatory development periods [66,67,106,107,108].

Nowadays, 3D bioprinting helps laboratories increase organoid production at a speed of thousands per hour in order to test the effects of infectious diseases and drug therapies. With the purpose of recreating a native organ’s environment, scientists also use devices called organs-on-a-chip [109]. The next scientific stage is to develop multiple organs-on-a-chip that are combined together in a system end-to-end. For example, a cancer drug can be pumped through a tumour-on-a-chip that is connected to a healthy tissue-on-a-chip to find out if a treatment dose shrinks the tumour without damaging healthy tissue [110]. 

Some scientists envisage smart tissue factories with high-tech, sterile clean rooms. Customized 3D bioprinted tissues and organ-on-a-chip devices could be produced by pharmaceutical companies, thus helping to achieve very rapid testing with greater sophistication and precision. Oncologists could also send an individual’s cancer tissues and drug formulations to the factory to be tested on 3D manufactured chip devices [52].

One should not ignore the fact that there is a certain risk in the use of animals for the production of vaccines, antibiotics, etc., which are utilized in diagnostics as well as for treatment. There is evidence that people have been harmed in the clinical testing of drugs that were considered safe by animal studies. The reasons for this can be different species genotypes, individual immune responses in different diseases, etc. [68].

Often, animal models are used to comply with regulatory agencies of efficacy and safety by in vivo preclinical testing of human therapies. Despite the fact that their usefulness cannot be contended (e.g., wound healing therapies), the truth is that in most instances the lack of genetic, molecular, and physiological relevance to human clinical conditions severely hinders their success in human predictability [69].

In this regard, biofabrication models deliver more precise human biological structures as they are able to recreate the functional organization of human tissues. If we manage to solve these obstructions and advancement continues at the same rate as in the past decade, it is highly possible that we will soon see the effective integration of bioprinting with stem cell reprogramming and gene editing technologies. This will definitely open the door to a new age of personalized medicine tissue models [111].

### 3.3. Surgery Planning and Medical Training for Young Doctors

Medical training and doctor–patient interactions have widely used medical models or “phantoms”. They are a preferred tool for surgical planning, medical computational models, algorithm verification and validation, and medical device development. These new applications require highly accurate, patient-specific, tissue-mimicking medical phantoms that not only closely reproduce the geometric structures of human organs but also possess the characteristics and functions of the organ structure [70,112]. 3D-printed medical models and phantoms produced on the basis of CT, MRI, or echocardiography data present the superiority of tactile feedback, direct manipulation, and a comprehensive knowledge of a patient’s anatomy and fundamental pathologies [71]. 

3D-printed medical models can be of assistance to surgeries and also facilitate them, and they can reduce the cycle times of medical procedures. 3D printing can also be beneficial when it comes to patient education, medical training, and surgical planning. Specialties such as orthopedic surgery, plastic surgery, and even cardiovascular surgery utilize 3D and bioprinting trying to adopt and acknowledge their integration into medical practice [113].

Currently, 3D printed models are widely utilized in surgery planning.

Recent advance in stem cell technology is expected to compel a paradigm transformation in regenerative medicine and disease modelling. The combination of induced pluripotent stem cell (iPSC) technology with advanced bioprinting systems can eventually provide a new generation of disease models and tissue structures that could be used for developing and testing personalized therapies with better efficacy and less cost [72,114]. This is why bioprinting will be of key importance for disease modelling and in fabricating multilayer native-like arrangements of cells for tissue reconstruction [63]. 

Digital fabrication technologies are not novel methods and means for the educational process. They can assist in learning, developing skills, inspiring creativity, improving attitudes and activity towards practicals and workshops, and at the same time they might stimulate interest and engagement [16]. Today, it is easier to obtain images of bodies for 3D printing than to receive a donated whole body [115].

In recent years, there has been growing interest in the use of digitalisation in medical education and training. 3D printing also helps surgical teams in planning a surgery, thereby ensuring a clear picture of the entire surgical approach with all of its steps. However, this advantage is not only limited to patient education or surgery planning as it can also be transferred to areas of medical training like anatomy and vascular access [73,74]. 

One of the major advantages is that students can make an examination without causing harm to patients [75,116]. In addition, it can be used for storage of patient cases affected by rare pathologies for medical education. It can provide opportunities to train doctors in specific rare pathologies and hard-to-treat diseases [117,118]. Printed models can reduce the need for human body parts, and alleviate ethical and legal concerns regarding the use of corpses for education purposes [119]. 

In various studies, the authors conclude that the 3D printed model may improve students’ understanding of medicine, pharmacy, and dental medicine, and it may enhance student satisfaction in their education [76,77,78,79,80,81,120,121]. In order to meet the ever-increasing demands for high-quality medical education, it is necessary to introduce new technologies, including 3D (bio)printing, as early in the training process as possible in order that medical professionals can be better prepared for their future activity.

## 4. Perspectives

However, bioprinting finds the greatest difficulties in the fabrication of secretory organs. The anatomical substance can be relatively easily set and formatted to comply with the individual characteristics of the recipient, but the function is difficult to replicate in, for instance, secretory organs like the pancreas, the liver, etc. On the other hand, another challenge in the future will be the fabrication of 3D (bio)printed organs or tissues for children that should grow symmetrically with the child’s body. This sets an age limit for application, or, in the case of a life-saving emergency, periodic organ replacement will be required [104]. So, as is the case with any new technology, 3D printing has brought many benefits and opportunities in the field of medicine.

## 5. Study Strengths and Limitations

The present review used the PRISMA checklist to evaluate and select the included scientific articles. Numerous systematic reviews were also published from 2017 to 2022, and the list of articles included in this study may be incomplete. However, we are unaware of any other similar systematic review summarizing the benefits of bioprinting in medicine. The main limitation was related to the methodology of the studies and differences in study design, material, and methods. In addition, this review does not cover the socio-ethical and economic benefits of 3D printing applications nor the aspects related to the quality of life. At this stage, more studies need to be accumulated on the effects and benefits, both in terms of the consequences after the application of bioprinting and the quality of life of patients who have received bioprinted products. Studies in this area are episodic, and it will take time to track both the benefits and the negatives. Nonetheless, this would significantly enhance future systematic reviews in 3D bioprinting.

## 6. Conclusions

3D (bio)printing advantages and opportunities were selected and explored as fabrication methods in the current review. The introduction of additive manufacturing and bioprinting has the potential to largely solve the problem of providing organs for transplantation. 3D printing and bioprinting are promising technological progress in various fields of medicine, and this possibility is rising. Indeed, the possibility of personalizing and adapting pharmacological therapies to the individual needs of each patient through 3D printing technologies is an indisputable advantage, which will have a huge impact on the development of personalized medicine [122,123]. However, as to 3D bioprinting of whole organs, the technology is still in its early stage and has a long way to go. 

The development of printing technologies in the field of healthcare represents a huge opportunity for the creation of more precise medicines, unified and faster production of different types of tissues and organs, etc. With technological advancements, 3D-printed implantable organs and tissues will be available for transplantation in the near future, reducing the in-need patient lists while also significantly increasing the number of human lives saved. At the same time, 3D (bio)printing creates the best conditions for improving modern personalized medicine and improving the quality of life.

More and more research organizations and government regulatory agencies are acknowledging that alternative methods may replace animal testing and improve the flow and safety of new therapeutics for human use. Great potential rests in bioprinting technologies with regards to eliminating tests on animals in pharmaceutical industries and delivering patient-specific drug testing. The latest research shows that bioprinted structures are much more manageable and reproducible than animal models, can provide significantly more reliable data, and may lead to a decrease or even the cessation of animal testing. In addition, 3D printed models can potentially assist medical education and considerably improve the practical skills of students and young doctors. Last but not least, we must note that the huge financial health costs will be notably reduced with the introduction of these technologies.

Based on the presented significant opportunities, it can be concluded that bioprinting can be considered a revolution in the development of personalized medicine, and it may soon become the gold standard of future healthcare.

## Figures and Tables

**Figure 1 bioengineering-10-00287-f001:**
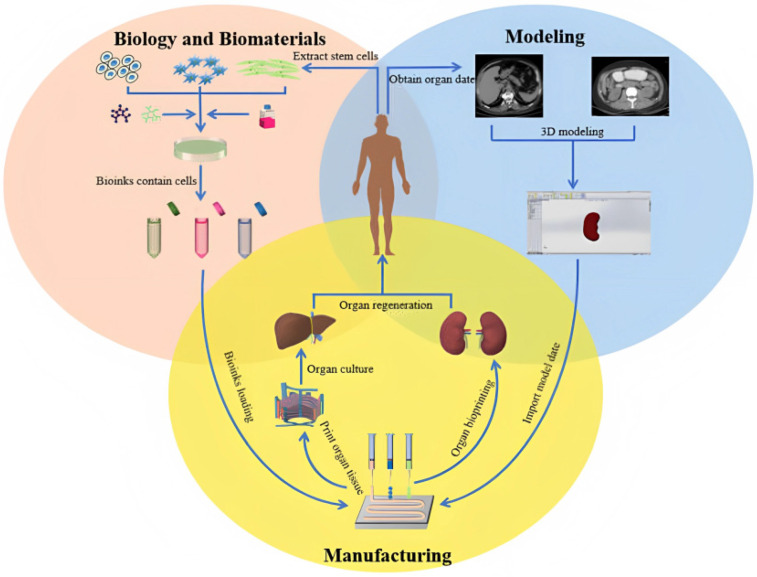
Organ 3D printing process. Reproduced under open access from [19] MDPI, 2021.

**Figure 2 bioengineering-10-00287-f002:**
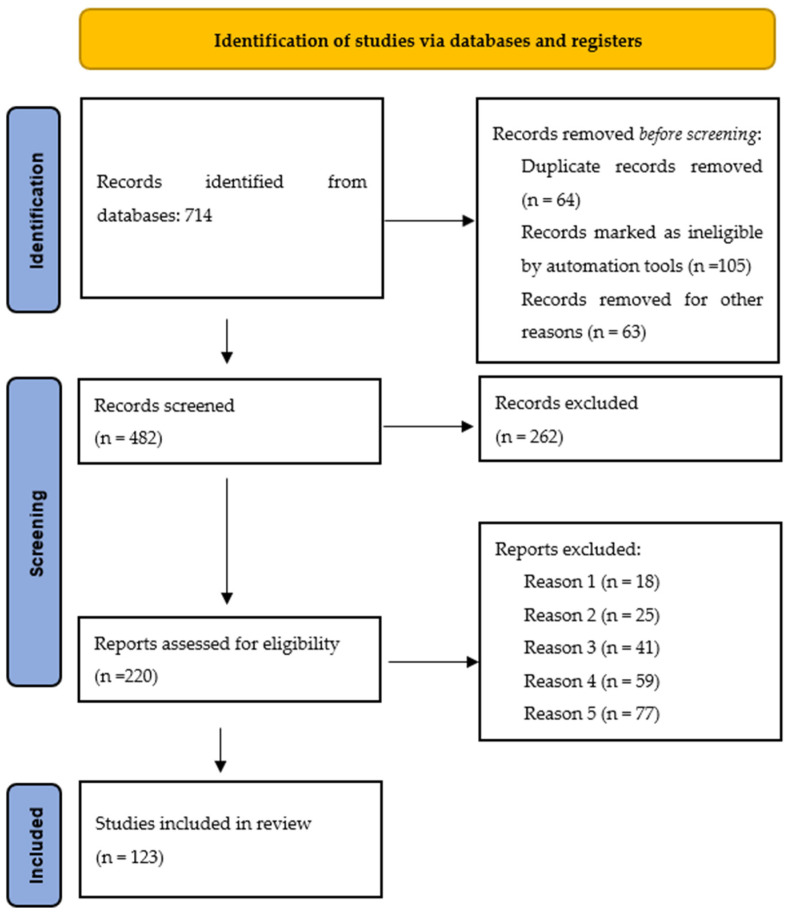
PRISMA flow chart for the selection process of articles.

**Table 1 bioengineering-10-00287-t001:** Summary of the bioprinting benefits.

Key Benefit/Topic	Area of Application/Significance	Authors
**Transplant organ shortage**	Organ transplantation	Bastani 2020 [41]
	Organ and tissue donation	Oliver 2023 [42]
	Blockchain-specific approach for the prevention or monitoring of organ trafficking	Patil et al., 2023 [43]
	the challenges and opportunities in tackling the potential reduction in organ donations	Mills 2020 [44]
	3D Bioprinting as an alternative to traditional organ transplantation	Iram et al., 2019 [45]
**Overcoming gender differences in transplantation**	Sex differences in transplantation	Momper et al., 2017 [46]
	Post-transplant infections	Vnucak et al., 2022 [47]
	Age-dependent sex differences after transplantation	Vinson et al., 2022 [48]
	Sex-based discrepancies between donors and recipients	Serrano et al., 2022 [49]
	Sex disparities in post-transplant survival	Gabbay et al., 2022 [50]
**Probability reduction of transplant rejection**	Alternatives to Organ Replacement	Paisarntanawat 2022 [51]
	Multidisciplinary teams seek to create living human organs	Tibbetts 2021 [52]
	Applications of 3D bioprinting technology for tissue engineering	Yu et al., 2020 [53]
	Skin bioprinting: the future of burn wound reconstruction	Varkey et al., 2019 [54]
	Orthopedics	Zheng et al., 2019 [55]
	Traumatic fractures	Yang et al., 2021 [56]
	Implants	Voelker 2021 [57]
	Heart tissue regeneration	Shahzadi et al., 2021 [58]
	Skin regeneration, repair, and reconstruction	Kamolz 2022 [59]
**Removal of congenital defects of various tissues and organs**	Orthopedics	Zheng et al., 2019 [55]
	Components of the human heart	Lee et al., 2019 [60]
	Tracheal reconstruction	Frejo and Grande 2019 [61]
	Bioprinting Skin	Kang et al., 2022 [61]
	Lung and tracheal tissue engineering	Mahfouzi et al., 2021 [62]
	Meniscus regeneration	Stocco et al., 2022 [63]
	Bone Regeneration	Wang et al., 2021 [64]
**Drug research and development**	Personalized cancer treatment	Mao et al., 2020 [65]
	Production of drug and cell-based systems	Bom et al., 2021 [66]
	Tissue constructs for disease modelling and drug testing	Moldovan 2021 [67]
	3D bioprinting as alternatives to animal research	Van Norman 2019 [68]
	Cancer Research	Jackson and Thomas 2017 [69]
**Surgery Planning and Medical training for young doctors**	Functional Structures for Medical Phantoms	Wang et al., 2017 [70]
	Comparison between 3D printout models and 3D-rendered images	Zheng et al., 2018 [71]
	Healthy and diseased models	Gu et al., 2017 [72]
	Preoperative Surgical Planning	Tejo-Otero et al., 2022 [73]
	Preoperative Surgical Planning	Segaran et al., 2021 [74]
	Phantom for Fine-Needle Aspiration Cytology	Baba et al., 2017 [75]
	Teaching and learning of bone spatial anatomy	Wu et al., 2018 [76]
	Pharmacy education	Lee and Lee 2021 [77]
	Teaching and learning in anatomy	Losco et al., 2017 [78]
	Cardiac surgery	Milano et al., 2019 [79]
	Dentistry and maxillofacial surgery	Khorsandi et al., 2021 [80]
	3D printed simulation models	Kröger et al., 2017 [81]

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
