# Peer review of "(Bio)printing in Personalized Medicine—Opportunities and Potential Benefits"

_bioengineering, 2023, doi:10.3390/bioengineering10030287_

Round 1

Reviewer 1 Report

Overall the work is interesting and provides a new angle for researches. This is why I believe should be published in the journal. However, there are few comments that need to be addressed before publication.

  1. The keywords utilize for the research are not clear. Are you using them separately in your research or una. Combined way? This point  regarding the Mesh terms should be addressed.
  2. I think the style of writing should be improved. Paragraphs are consisting of 1 sentence sometimes which si nto grammatically correct.
  3. The review is really lacking images to depict in a shematic way the different sections of the paper.
  4. Tables summarising the information are missing. I recommend the authors to collect data into tablets for the reader to have an idea just at glance.
  5. The discussion is poor I think you should open up the concept slightly to other fields, or just briefly discussed them, what about biopringing with cells to repare tissue? Have you considered to briefly mentioned the difficulties and challenges for printing each of the structures proposed? Have a look to these papers: Personalised 3D printed medicines: which techniques and polymers are more successful? And 3D printing technologies in personalized medicine, nanomedicines, and biopharmaceuticals

Author Response

Overall the work is interesting and provides a new angle for researches. This is why I believe should be published in the journal. However, there are few comments that need to be addressed before publication.

  1. The keywords utilize for the research are not clear. Are you using them separately in your research or una. Combined way? This point regarding the Mesh terms should be addressed.

The search strategy was described in details: lines 161-167

     2. I think the style of writing should be improved. Paragraphs are consisting of 1 sentence sometimes which si nto grammatically correct.

The style of the manuscript was improved and the grammar was double checked and corrected.

     3. The review is really lacking images to depict in a shematic way the different sections of the paper.

The appropriate figure was added in the introduction section (Figure 1): lines 91-94

     4. Tables summarising the information are missing. I recommend the authors to collect data into tablets for the reader to have an idea just at glance.

Information was summarized in Table1 : line 226

      5. The discussion is poor I think you should open up the concept slightly to other fields, or just briefly discussed them, what about biopringing with cells to repare tissue? Have you considered to briefly mentioned the difficulties and challenges for printing each of the structures proposed? Have a look to these papers: Personalised 3D printed medicines: which techniques and polymers are more successful? And 3D printing technologies in personalized medicine, nanomedicines, and biopharmaceuticals

The discussion and perspectives as well as strengths and limitations of our study were described in details (lines: 469 – 500). Recommended articles were included too. In the future we will continue our scientific research in this fast-developing field.

Reviewer 2 Report

There are some weaknesses through the manuscript which need improvement. Therefore, the submitted manuscript cannot be accepted for publication in this form, but it has a chance of acceptance after a major revision. My comments and suggestions are as follows:

1- Abstract gives information on the main feature of the performed study, but a couple of sentences about the highlights of the review must be added.

2- Authors must clarify necessity of the performed research. Objectives of the study, must be clearly mentioned in introduction.

3- There are articles on reinforced composites which are relevant to discussion. It is recommended to cite the articles for completeness: (a) https://doi.org/10.1016/j.sna.2020.112105 and (b) https://doi.org/10.1016/j.sna.2023.114154 and other research works.

4- Authors must discuss limitations and strength of their review in details.

5- It seems the manuscript is prepared without care. For example, identical affiliations are used with different numbers (3-6).

6- It is a review paper and I am wondering there is no figures from the reviewed papers. Authors must check similar review papers to learn to put figures from reviewed research works.

7- Advantages and limitations for each applications must be discussed in details. Adding tables is a good idea.

8- In its language layer, the manuscript should be considered for English language editing. There are sentences which have to be rewritten. All mathematical formula and indices must be checked.

9- The current version of conclusion is too short. The conclusion must be more than just a summary of the manuscript. List of references must be updated based on the proposed papers. Please provide all changes by red color in the revised version.

Author Response

There are some weaknesses through the manuscript which need improvement. Therefore, the submitted manuscript cannot be accepted for publication in this form, but it has a chance of acceptance after a major revision. My comments and suggestions are as follows:

1. Abstract gives information on the main feature of the performed study, but a couple of sentences about the highlights of the review must be added.

The abstract was updated: lines 30-38

2. Authors must clarify the necessity of the performed research. Objectives of the study, must be clearly mentioned in introduction.

A new paragraph was added to clarify the goal of our study: lines 145-152

3. There are articles on reinforced composites which are relevant to discussion. It is recommended to cite the articles for completeness: (a) https://doi.org/10.1016/j.sna.2020.112105 and (b) https://doi.org/10.1016/j.sna.2023.114154 and other research works.

The articles were cited in the introduction for completeness: lines 62-65

4. Authors must discuss limitations and strength of their review in details.

Study strengths and limitations section was added: lines 479-491

5. It seems the manuscript is prepared without care. For example, identical affiliations are used with different numbers (3-6).

The affiliations were corrected according to the MDPI guidelines.

6. It is a review paper and I am wondering there is no figures from the reviewed papers. Authors must check similar review papers to learn to put figures from reviewed research works.

The appropriate figure was added in the introduction section (Figure 1): lines 91-94

7. Advantages and limitations for each applications must be discussed in details. Adding tables is a good idea.

The table with the details was included (Table 1): line 226

8. In its language layer, the manuscript should be considered for English language editing. There are sentences which have to be rewritten. All mathematical formula and indices must be checked.

The English Language and style of the manuscript were improved.

9. The current version of conclusion is too short. The conclusion must be more than just a summary of the manuscript. List of references must be updated based on the proposed papers. Please provide all changes by red color in the revised version.

The conclusion ( lines 492- 522) was updated.

The references were corrected according to the journal guidelines.

Round 2

Reviewer 2 Report

The paper has been improved and corresponding modifications have been conducted. In my opinion, the current version can be considered for publication.